Phylogeny and divergence times of suckers (Cypriniformes: Catostomidae) inferred from Bayesian total-evidence analyses of molecules, morphology, and fossils

http://orcid.org/0000-0001-6737-8380 Bagley Justin C. 1 2 3 jcbagley@vcu.edu
Mayden Richard L. 4
http://orcid.org/0000-0003-3215-2167 Harris Phillip M. 5
1 Department of Biology, Virginia Commonwealth University , Richmond, VA , USA
2 Departamento de Zoologia, Universidade de Brasília , Brasília, Distrito Federal , Brazil
3 Departamento de Zoologia e Botânica, IBiLCE, Universidade Estadual Paulista , São José do Rio Preto, São Paulo , Brazil
4 Department of Biology, Saint Louis University , St. Louis, MO , USA
5 Department of Biological Sciences, The University of Alabama , Tuscaloosa, AL , USA
De Baets Kenneth
Electronic publication date: 2018 Jul 4
Publication date: 2018
Volume: 6
Electronic Location ID: e5168
Received 2018 May 2; Accepted 2018 Jun 15
Copyright: © 2018 Bagley et al.
Copyright year: 2018
Copyright holder: Bagley et al.
License: This is an open access article distributed under the terms of the Creative Commons Attribution License, which permits unrestricted use, distribution, reproduction and adaptation in any medium and for any purpose provided that it is properly attributed. For attribution, the original author(s), title, publication source (PeerJ) and either DOI or URL of the article must be cited.
License URL: https://creativecommons.org/licenses/by/4.0/

Keywords: Molecular phylogenetics, Catostomidae, Fossilized birth-death process, Divergence time estimation, Relaxed molecular clock, Phylogenetic informativeness profiles

Funding: U.S. National Science Foundation (NSF) EF-0431326 and EF-0431263 Brazilian Ciência Sem Fronteiras postdoctoral fellowship from CNPq Processo 314724/2014-1 This research was funded by U.S. National Science Foundation (NSF) grants EF-0431326 to Richard L. Mayden and EF-0431263 to Phillip M. Harris (Cypriniformes Tree of Life Project), and Justin C. Bagley received stipend support from a Brazilian Ciência Sem Fronteiras postdoctoral fellowship from CNPq (Processo 314724/2014-1). The funders had no role in study design, data collection and analysis, decision to publish, or preparation of the manuscript.

==============================
Catostomidae (“suckers”) is a diverse (76 species) and broadly distributed family of Holarctic freshwater fishes with a rich fossil record and a considerable number (∼35%) of threatened and imperiled species. We integrate DNA sequences (three mitochondrial genes, three nuclear genes), morphological data, and fossil information to infer sucker phylogenetic relationships and divergence times using Bayesian “total-evidence” methods, and then test hypotheses about the temporal diversification of the group. Our analyses resolved many nodes within subfamilies and clarified Catostominae relationships to be of the form ((Thoburniini, Moxostomatini), (Erimyzonini, Catostomini)). Patterns of subfamily relationships were incongruent, but mainly supported two placements of the Myxocyprininae; distinguishing these using Bayes factors lent strongest support to a model with Myxocyprininae sister to all remaining sucker lineages. We improved our Bayesian total-evidence dating analysis by excluding problematic characters, using a clock-partitioning scheme identified by Bayesian model selection, and employing a fossilized birth-death tree prior accommodating morphological data and fossils. The resulting chronogram showed that suckers evolved since the Late Cretaceous–Eocene, and that the Catostomini and Moxostomatini clades have accumulated species diversity since the early to mid-Miocene. These results agree with the fossil record and confirm previous hypotheses about dates for the origins of Catostomide and catostomine diversification, but reject previous molecular hypotheses about the timing of divergence of ictiobines, and between Asian–North American lineages. Overall, our findings from a synthesis of multiple data types enhance understanding of the phylogenetic relationships, taxonomic classification, and temporal diversification of suckers, while also highlighting practical methods for improving Bayesian divergence dating models by coupling phylogenetic informativeness profiling with relaxed-clock partitioning.

Introduction

“Suckers” in the family Catostomidae (Cypriniformes) form a large family of Holarctic freshwater fishes with 76 extant species in 14 genera native to North America and Asia (Harris, Hubbard & Sandel, 2014). Seventy-five species from 13 genera occur in North America (Harris, Hubbard & Sandel, 2014; Nelson et al., 2004; Unmack et al., 2014), where they constitute the third largest freshwater fish clade, after darters (Etheostomatinae) and minnows (Cyprinidae), and ∼8% of the continental ichthyofauna (Warren et al., 2000). Myxocyprinus asiaticus is endemic to eastern China and Catostomus catostomus is the only extant trans-Pacific species (Harris, Hubbard & Sandel, 2014). Suckers are an ancient group whose fossil record spans the Cenozoic, from the early Eocene onwards (Cavender, 1986; Smith, 1992; Smith et al., 2002; Appendix S1). Around 35% of the taxa (26 to >35 species or genetic lineages) are endangered, threatened, or of special conservation concern (Harris, Hubbard & Sandel, 2014; Nelson et al., 2004; Warren et al., 2000).

Geographical distributions of suckers and their taxonomy and relationships have attracted the interest of systematists and biogeographers for over 150 years (Agassiz, 1854). Pre-1900 systematics and taxonomy studies dealt with species descriptions and higher-level classifications of the group (reviewed by Harris, Hubbard & Sandel, 2014; Smith, 1992). Subsequently, key papers on sucker classification designated genera, subgenera, and tribes (Hubbs, 1930; Robins & Raney, 1956) and contributed a pre-cladistics phylogeny (Miller, 1959), as well as the first phylogeny for species in the tribe Moxostomatini (Jenkins, 1970). The findings of post-systematics studies of sucker interrelationships (last 40 years), which were based on a variety of different data types, are summarized in Table 1 and Fig. 1. These studies supported the recognition of four subfamilies (Myxocyprininae, Cycleptinae, Ictiobinae, and Catostominae) and tribes (Catostomini, Erimyzonini, Moxostomatini, Thoburniini) plus the modern allocation of genera to these groups (Harris & Mayden, 2001; Harris et al., 2002). Most phylogenetic analyses of suckers to date have analyzed DNA sequence data from increasing numbers of mitochondrial DNA (mtDNA) or nuclear DNA (nDNA) genes. Recent advances include Clements, Bart & Hurley’s (2012) study of phylogenetic relationships of Moxostomatini using the first dataset based on two loci—mtDNA cytb and nuclear growth hormone intron (GHI) sequences. Also, Unmack et al. (2014) studied the phylogeny and biogeography of Pantosteus, a former subgenus of Catostomus (Harris & Mayden, 2001; Harris et al., 2002; Smith, 1966) using independent morphological and mtDNA genetic analyses. Overall, while the inclusion of additional taxa and sequence data in recent analyses has yielded novel insights on sucker relationships, a consensus among hypotheses regarding relationships of higher-level species groups, i.e., subfamilies, has yet to emerge (Table 1; Fig. 1). Moreover, although time-calibrated phylogenies based on comprehensive tip sampling are necessary to understand the tempo and mode of diversification of groups of freshwater fishes (Near et al., 2011, 2012), such a tool has yet to be inferred for suckers. This leaves workers at a distinct disadvantage in considering the macroevolutionary, ecological, or conservation trends of suckers in a phylogenetic context.

Table 1 Summary of post-systematic phylogenetic and taxonomic studies of suckers in the family Catostomidae.

Study	n	Data type	Analysis	Supported clades/relationships	
Ferris & Whitt (1978)	30	20 Isozyme loci (loss of duplicate gene expression)	Wagner	Ictiobinae + (Cycleptinae + Catostominae), three catostomin tribes	
Smith (1992)	64	157 Morphological, biochemical and early life-history transformation series	MP	Ictiobinae + (Cycleptinae + Catostominae), Moxostoma in paraphyletic grade with “Scartomyzon” within Moxostomatini; also trichotomy of M. ariommum + Thoburnia + Hypentelium	
Harris & Mayden (2001)	16	mtDNA SSU and LSU rDNA sequences	MP	Four monophyletic subfamilies, Thoburniini resurrected and includes Hypentelium; Catostomini + Erimyzonini and Moxostomatini + Thoburniini	
Harris et al. (2002)	50	mtDNA cytochrome b (cytb) gene	MP and ML	Four monophyletic subfamilies, provisional Thoburniini; “Scartomyzon” as a junior synonym of Moxostoma	
Sun et al. (2007)	17	mtDNA cytb gene	UPGMA	Monophyletic Ictiobinae and Catostominae; Moxostomatini + Catostomini	
Doosey et al. (2010)	60	mtDNA NADH subunit 4 and 5 (ND4/5) sequences	ML	((Cycleptinae, (Myxocyprininae, Ictiobinae)), Catostominae); also among catostomins: Erimyzonini + (Moxostomatini + Thoburniini)	
Chen & Mayden (2012)	67	Interphotoreceptor retinoid-binding protein (IRBP) sequences	ML	Supported previous relationships including monophyly of subfamilies and tribes except Thoburniini	
Clements, Bart & Hurley (2012)	45	Growth hormone intron (GHI)	ML	Para-/polyphyletic Moxostoma and Scartomyzon	
Unmack et al. (2014)	24	Morphology, plus cytb and other mtDNA gene sequences	MP, ML, and Bayesian	Monophyletic Pantosteus within Catostomini, except for Catostomus (P.) columbianus (potential hybrid origin)	
Note:

ML, maximum likelihood; MP, maximum parsimony; n, sample size, including ingroup and outgroup taxa or lineages; UPGMA, unweighted pair-group method with arithmetic mean; Wagner, Wagner method for inferring “most parsimonious” tree.

Figure 1 Six alternative hypotheses of phylogenetic relationships among Catostomidae subfamilies and tribes.

These results were based on previous studies of electrophoretic data (A); morphological, behavioral, and developmental characters (B); and DNA sequence data (C–F).

In this study, we use “total-evidence” analyses of multilocus sequence data from three mtDNA genes and three nuclear genes representing the most comprehensive sampling of suckers to date, combined with available morphological and fossil data, to infer the phylogenetic relationships and divergence times of suckers in a Bayesian framework. We discuss the phylogenetic and taxonomic implications of our results in light of previous phylogenetic studies of suckers. We apply Bayesian divergence time methods to the data (Drummond et al., 2012) and take advantage of rich information on the age and distributions of sucker fossils (Cavender, 1986; Smith et al., 2002; Appendix S1) to improve divergence time estimation by incorporating extant and fossil sampling in a “fossilized birth-death process” tree prior (Stadler, 2010; Gavryushkina et al., 2014; Heath, Huelsenbeck & Stadler, 2014). Moreover, by coupling assessments of the phylogenetic signal of different data subsets with evaluation of clock-partitioning strategies, we were able to avoid potentially confounding effects of problematic characters on our divergence time inferences. The resulting time-calibrated phylogeny is then used to test several hypotheses about the temporal diversification of suckers. Namely, we test the hypotheses (H1) that Asian Myxocyprininae and North American suckers (Cycleptinae) diverged since ∼14 million years ago (Ma) in the mid-Miocene, and (H2) that the initial divergence of Ictiobinae lineages occurred since ∼10 Ma in the late Miocene, as indicated by mtDNA cytb gene divergences (Sun et al., 2007). We also tested the hypotheses (H3) that the diversification of lineages within Catostominae followed ∼20 Ma in the Miocene (Smith, 1992; Sun et al., 2007), and (H4) that Catostomidae species have diversified from a tetraploid ancestor since ∼50 Ma in the early Eocene (Uyeno & Smith, 1972). Overall, by yielding a new phylogeny of suckers and divergence dates for their most recent common ancestors (MRCAs), our study sheds light on the interrelationships, taxonomic classification, and tempo of speciation in a diverse and threatened clade of Holarctic freshwater fishes.

Materials and Methods

Molecular taxon sampling, laboratory methods, and sequence alignment

We obtained and sequenced mtDNA and nDNA genes from tissue samples of 121 sucker specimens from throughout the geographical range of the family. Samples were provided by ichthyological collections or colleagues (see Acknowledgements), or were already in-hand at the beginning of the study. We sequenced taxa representing all sucker genera and all Catostominae species except for †Moxostoma lacerum, an historically extinct taxon last sampled from the Mississippi River Basin in 1893 (NatureServe, 2013), and Chasmistes cujus, an endangered species (NatureServe, 2014) for which it was difficult to obtain samples (Table 2). To complement our sampling and add data from more unlinked loci to our datasets, we obtained additional sequences from GenBank as described below. Overall, we obtained genetic sequences for the most comprehensive taxonomic sampling of the family to date, including 78 species/lineages representing all 14 extant genera, including four “candidate species” within Moxostomatini (Table 2). Based on recent hypotheses of relationships of suckers and cypriniform fishes (Mayden et al., 2008; Saitoh et al., 2006; Smith, 1992), we included DNA sequences from five outgroup taxa in our datasets: Cyprinus carpio (Cyprinidae); Gyrinocheilus aymonieri (Gyrinocheilidae); and Cobitis striata, Chromobotia macracanthus, and Leptobotia mantschurica (Cobitidae). We follow the taxonomy of subfamilies listed in Harris & Mayden (2001) and Harris et al. (2002).

Table 2 List of sequences used in the present study including museum/field numbers and GenBank accession numbers.

Taxona	Museum/field numbersb	cytb	ND2	cox1	IRBP	RPS7	GHI	
Order Cypriniformes								
 Family Cyprinidae								
  Subfamily Cyprininae								
   Cyprinus carpio	CToL 1751	AP009047	AP009047	EU524006	FJ197101	DQ163924	FJ265047	
 Family Gyrinocheilidae								
   Gyrinocheilus aymonieri	CToL 612	NC008672	NC008672	JF915620	JX470019	–	FJ265031	
 Family Cobitidae						–	–	
  Subfamily Cobitinae								
   Cobitis striata	CToL 230	NC004695	NC004695	–	Mayden et al. (2009; K. Saitoh, unpublished data)	–	–	
  Subfamily Botinae								
   Chromobotia macracanthus	CToL 217	AC024175	AC024175	KF738207	FJ197086	–	–	
   Leptobotia mantschurica	–	AB242170	AB242170	AB242170	FJ197087	–	FJ265035	
 Family Catostomidae								
  Subfamily Myxocyprininae								
   Myxocyprinus asiaticus 1	–	NC006401	NC006401	AY526869	–	–	–	
   M. asiaticus 2	–	AY526869	AY526869	–	–	–	–	
   M. asiaticus 3	UAIC 11698.01	JX488760	JX488826	–	JX488937	This study	FJ265052	
  Subfamily Ictiobinae								
   Carpiodes carpio 1	–	AY366087	AY366087	–	–	–	–	
   Car. carpio 2	UAIC 11219.08	AF454867	JX488827	JN024866	–	–	JF837387	
   Car. cyprinus	TU 157.01	JX488761	JX488828	EU523924	JX488938	–	GU937849	
   Car. Velifer	TU 108.16	JX488762	JX488829	JN024878	JX488939	–	JF837435	
   Ictiobus bubalus 1	TU 124.07	JX488763	JX488830	KF929996	JX488940	–	JF799533	
   I. bubalus 2	TU 120.01	JX488764	JX488831	–	JX488941	–	–	
   I. bubalus 3	TU 244.03	JX488765	JX488832	–	JX488942	–	–	
   I. cyprinellus 1	TU 101.05	JX488766	JX488833	EU524687	–	–	–	
   I. cyprinellus 2	TU 107.15	JX488767	JX488834	–	JX488943	–	GU937840	
   I. niger 1	TU 121.01	JX488768	JX488835	EU524107	JX488944	–	FJ226251	
   I. niger 2	TU 124.06	JX488769	JX488836	–	JX488945	–	–	
  Subfamily Cycleptinae						–	–	
   Cycleptus elongatus 1	–	NC008645	NC008645	–	–	–	–	
   Cyc. elongatus 2	UAIC 11371.01	AF454868	JX488837	KF929801	JX488946	–	FJ265028	
   Cyc. meridionalis	UNL 1-30	JX488770	JX488838	–	JX488947	–	–	
  Subfamily Catostominae								
   Tribe Catostomini								
    Catostomus ardens 1	USU-UBL	JX488771	JX488839	JN024886	JX488948	–	–	
    C. ardens 2	USU-USD	JX488772	JX488840	–	JX488949	–	–	
    C. bernardini Río Papogochic	BRK0507-2	JX488773	JX488841	EU668462	JX488950	–	–	
    C. bernardini Río Batopilas 1	JEB0504-2	JX488774	JX488842	–	JX488951	–	–	
    C. bernardini Río Batopilas 2	JEB0505-7	JX488775	JX488843	–	JX488952	–	–	
    C. bernardini Río Yecora	JEB0510-1	JX488776	JX488844	–	JX488953	–	–	
    C. cahita 1	BRK0511-6	JX488777	JX488845	EU668375	JX488954	This study	–	
    C. cahita 2	BRK0511-8	JX488778	JX488846	–	JX488955	This study	–	
    C. catostomus	–	–	–	W.J. Chen	W.J. Chen, unpublished data	–	–	
    C. catostomus	UAIC 11237.04	AF454871	JX488847	EU523925	–	–	GU937824	
    C. columbianus	OS 17548	JX488780	JX488849	–	JX488957	–	–	
    C. commersoni 1	–	NC008647	NC008647	–	–	This study	–	
    C. commersoni 2	UAIC 11156.03	JX488781	JX488850	EU523931	JX488958	This study	JF799535	
    C. fumeiventris 1	–	JX488784	JX488853	–	JX488961	–	–	
    C. fumeiventris 2	–	JX488785	JX488854	–	JX488962	–	–	
    C. insignis	MSB 49603-2	JX488786	JX488855	–	JX488963	–	–	
    C. insignis	MSB 49603-4	JX488787	JX488856	HQ556974	JX488964	This study	–	
    C. latipinnis 1	MSB 49601	JX488788	JX488857	–	JX488965	This study	–	
    C. latipinnis 2	MSB 49602	JX488789	JX488858	–	JX488966	–	–	
    C. leopoldi	JEB15-1	JX488790	JX488859	EU668394	JX488967	–	–	
    C. macrocheilus	OS 15886	JX488791	JX488860	EU523932	JX488968	–	–	
    C. microps	OS 17571-1	JX488792	JX488861	HQ557219	JX488969	This study	–	
    C. microps	OS 17571-2	JX488793	JX488862	–	JX488970	–	–	
    C. occidentalis humboldtianus 1	OS 15623-1	JX488795	JX488864	–	JX488972	–	–	
    C. occidentalis humboldtianus 2	OS 15623-2	JX488796	JX488865	–	JX488973	–	–	
    C. occidentalis lacusanserinus	OS 1943BB	JX488797	JX488866	–	JX488974	–	–	
    C. occidentalis mniotiltus 1	UAIC 13448.01-2	JX488798	JX488867	–	JX488975	–	–	
    C. occidentalis mniotiltus 2	UAIC 13448.01-4	JX488799	JX488868	–	JX488976	–	–	
    C. occidentalis occidentalis	UAIC 11546.02-1	JX488800	JX488869	KF558280	JX488977	This study	–	
    C. rimiculus	OS 15908	AF454875	JX488874	–	JX488982	–	–	
    C. snyderi	OS 15900	JX488806	JX488876	–	JX488984	–	–	
    C. tahoensis 1	BuffaloCrk4	JX488807	JX488877	JN024954	JX488985	This study	–	
    C. tahoensis 2	SmokeCrk1	JX488808	JX488878	–	JX488986	–	–	
    C. tsiltcoosensis 1	OS X113	JX488809	JX488879	–	JX488987	–	–	
    C. tsiltcoosensis 2	OS X114	JX488810	JX488880	–	JX488988	–	–	
    C. warnerensis 1	OS 14251-2	JX488814	JX488884	–	JX488992	This study	–	
    C. warnerensis 2	OS 14251-3	JX488815	JX488885	–	JX488993	–	–	
    C. wigginsi	JEB0511-1	JX488816	JX488886	EU668440	JX488994	–	–	
    Catostomus sp. Coquille River	OS X130	JX488811	JX488881	–	JX488989	–	–	
    Catostomus sp. Wall Canyon 1	OS X61	JX488812	JX488882	–	JX488990	–	–	
    Catostomus sp. Wall Canyon 2	OS X67	JX488813	JX488883	–	JX488991	–	–	
    Chasmistes brevirostris	OS 15963	JX488817	JX488887	–	JX488995	–	GU937825	
    Ch. liorus mictus 1	BYU 56945	JX488818	JX488888	–	JX488996	–	–	
    Ch. liorus mictus 2	BYU 56946	JX488819	JX488889	–	JX488997	–	–	
    Deltistes luxatus	OS 15922	AF454870	JX488890	–	JX488998	–	GU937831	
    Pantosteus clarkii	MSB 49600	JX488779	JX488848	HQ556940	JX488956	–	–	
    P. discobolus discobolus	BYU 57986	JX488782	JX488851	–	JX488959	–	GU937830	
    P. discobolus yarrowi	UAIC 12776.01-2	JX488783	JX488852	–	JX488960	–	–	
    P. nebuliferus	HLB1484	JX488794	JX488863	EU668538	JX488971	–	–	
    P. platyrhynchus 1	BYU 58618	JX488801	JX488870	EU523934	JX488978	–	–	
    P. platyrhynchus 2	BYU 58621	JX488802	JX488871	–	JX488979	–	–	
    P. plebeius 1	DAN0521-10	JX488803	JX488872	EU668409	JX488980	–	GU937833	
    P. plebeius 2	DAN0526-01	JX488804	JX488873	–	JX488981	–	–	
    P. santaanae	UAIC 12384.01	JX488805	JX488875	JN024948	JX488983	–	–	
    Xyrauchen texanus 1	OS X97	JX488824	JX488935	HQ556969	JX489042	This study	–	
    X. texanus 2	OS X98	JX488825	JX488936	–	JX489043	–	–	
   Tribe Erimyzonini						–	–	
    Erimyzon oblongus 1	UAIC 11109.09	AF454876	JX488891	HQ579034	–	This study	–	
    Erimyzon oblongus 2	BRK05-52	–	–	–	JX488999	–	GU937837	
    E. sucetta 1	UAIC 12286.01	AF45478	JX488892	EU524567	–	–	–	
    E. sucetta 2	C306	–	–	–	JX489000	–	–	
    E. tenuis	UAIC 12370.04	AF454877	JX488893	JN025452	JX489001	This study	GU937838	
    Minytrema melanops 1	–	AB242116	AB242116	EU524839	–	–	FJ265050	
    Minytrema melanops 2	–	DQ536432	DQ536432	–	–	–	–	
    Minytrema melanops 3	UAIC 11141.01	AF454879	JX488897	–	–	–	–	
    Minytrema melanops 4	BRK056	–	–	–	JX489005	–	–	
   Tribe Moxostomatini						–	–	
    Moxostoma albidum 1	UAIC 12365.01	AF454901	JX488898	EU751969	JX489006	This study	JF799563	
    M. albidum 2	UAIC 13446.01	AF454902	JX488899	–	JX489007	This study	–	
    M. anisurum	UAIC 11606.02	AF454880	JX488900	EU524146	JX489008	This study	JF799537	
    M. ariommum	UAIC 12071.01	AF454903	JX488901	JN027267	JX489009	This study	JF799557	
    M. austrinum	UAIC 12375.01	AF454898	JX488902	–	JX489010	This study	JF799565	
    M. sp. cf. austrinum	UAIC 12373.01	AF454904	JX488903	–	JX489011	This study	JF799564	
    M. breviceps	UAIC 11314.08	AF454888	JX488904	JN027271	JX489012	This study	JF799542	
    M. carinatum 1	UAIC 11005.03	AF454883	JX488905	EU524147	JX489013	This study	JF799547	
    M. carinatum 2	PBB0023	JX488820	JX488906	–	JX489014	This study	–	
    M. cervinum	UAIC 11004.01	AF454906	JX488907	JN027274	JX489015	This study	JF799556	
    M. collapsum 1	UAIC 11007.03	AF454882	JX488908	JN027276	JX489016	This study	–	
    M. collapsum 2	UAIC 12376.05	AF454881	JX488909	–	JX489017	This study	–	
    M. congestum 1	UAIC 13506.05	AF522290	JX488910	JN027282	JX489018	This study	JF799560	
    M. congestum 2	UAIC 13508.05	AF522291	JX488911	–	JX489019	This study	–	
    M. congestum 3	UAIC 13512.05	AF522292	JX488912	–	JX489020	This study	–	
    M. duquesnei 1	UAIC 11310.09	AF454894	JX488913	EU524861	–	–	–	
    M. duquesnei 2	JFBM38581	AF454895	JX488914	–	JX489021	This study	JF799554	
    M. erythrurum 1	UAIC 12237.03	AF454886	JX488915	EU524867	JX489022	This study	JF799551	
    M. erythrurum 2	JFBM37043	AF454887	JX488916	–	JX489023	This study	–	
    M. hubbsi 1	NCSM 36440	AF522289	JX488917	EU524877	JX489024	This study	JF799544	
    M. hubbsi 2	06-19	JX488821	JX488918	–	JX489025	This study	–	
    M. lachneri	UAIC 12370.02	AF454900	JX488919	JN027292	JX489026	This study	JF799559	
    M. sp. cf. lachneri	UAIC 12462.03	AF454907	JX488920	–	JX489027	This study	–	
    M. macrolepidotum	UAIC 11221.10	AF454890	JX488921	EU524149	JX489028	This study	JF799539	
    M. sp. cf. macrolepidotum	UAIC 11643.01	AF454885	JX488922	–	JX489029	This study	F799541	
    M. mascotae	UAIC 12374.01	AF454899	JX488923	–	JX489030	This study	JF799566	
    M. pappillosum	UAIC 13462.01	AF454883	JX488924	JN027303	JX489031	This study	JF799538	
    M. poecilurum 1	–	NC008674	NC008674	–	–	–	–	
    M. poecilurum 2	UAIC 11442.01	AF454896	JX488925	HQ579038	JX489032	This study	JF799552	
    M. sp. cf. poecilurum	UAIC 12746.13	AF454897	JX488926	–	JX489033	This study	–	
    M. pisolabrum	UAIC 11154.05	AF454889	JX488927	HQ557206	JX489034	This study	–	
    M. robustum	UAIC 11916.01	AF454891	JX488928	JN027313	JX489035	This study	JF799546	
    M. rupiscartes	UAIC 12376.06	AF454905	JX488929	JN027315	JX489036	This study	JF799558	
    M. valenciennesi	JFBM36305	AF454893	JX488930	EU524150	JX489037	This study	JF799543	
   Tribe Thoburniini								
    Hypentelium etowanum	UAIC 12523.08	AF454908	JX488894	JN026830	JX489002	–	GU937836	
    H. nigricans 1	–	NC008676	NC008676	–	–	–	–	
    H. nigricans 2	UAIC 11138.02	AF454909	JX488895	EU524667	JX489003	This study	JF799571	
    H. roanokense	UAIC 13449.02	AF454910	JX488896	JN026848	JX489004	This study	JF799570	
    Thoburnia atripinnis	UAIC 13463.01	AF454911	JX488931	HQ937020	JX489038	This study	JF799569	
    T. hamiltoni 1	NCSM 45840-1	JX488822	JX488932	–	JX489039	–	JF799567	
    T. hamiltoni 2	NCSM 45840-3	JX488823	JX488933	–	JX489040	This study	–	
    T. rhothoeca	UAIC 11009.05	AF454912	JX488934	JN028432	JX489041	This study	JF799568	
Notes:

Sequences generated during this study are shown in bold. Dash symbol “–,” indicates no sequence was available.

a Taxonomic classification of suckers used here is based on Harris & Mayden (2001) and Harris et al. (2002).

b Institutional, project, and individual abbreviations: BRK, Bernard R. Kuhajda; CToL, Cypriniformes Tree of Life Project; DAN, David A. Neely; JEB, James E. Brooks; JFBM, (James Ford) Bell Museum of Natural History, University of Minnesota; HLB, Henry L. Bart, Jr.; MSB, Museum of Southwest Biology; PBB, Peter B. Berendzen; UAIC, University of Alabama Ichthyological Collection; USU, Utah State University; UNL, University of Nebraska Lincoln; OS, Oregon State University; BYU, Monte L. Bean Life Science Museum, Brigham Young University; Louis Bernatchez, Université Laval; NCSM, North Carolina State Museum of Natural Sciences; TU, Tulane University Biodiversity Research Institute (formerly Tulane University Museum of Natural History).

All molecular laboratory work for this project conducted in the R. L. Mayden laboratory was approved under Saint Louis University Institutional Animal Care and Use Committee (IACUC) protocol #2467. We extracted whole genomic DNA using QIAGEN DNeasy Tissue kits (Catalog No. 69506; QIAGEN, Valencia, CA, USA), or the CTAB method of Saghai-Maroof et al. (1984). We amplified and sequenced the mtDNA cytb gene using polymerase chain reaction (PCR) primers and parameters in Harris et al. (2002). We amplified and sequenced the mtDNA NADH subunit 2 (ND2) gene with primers 562 (5′-TAA GCT ATC GGG CCC ATA CC-3′) and 449 (5′-TGC TTA GGG CTT TGA AGG CTC-3′) from LGL Genetics, Bryan, TX, using the same PCR amplification parameters used for cytb. We also sequenced the first part of the nuclear interphotoreceptor retinoid binding protein (IRBP) gene using PCR primers and methods in Chen et al. (2008), and we sequenced nuclear ribosomal protein S7 intron 1 (RPS7) using PCR primers and parameters in Chow & Takeyama (1998). We purified double-stranded PCR products using columns or gel extraction kits (QIAGEN, Valencia, CA, USA). Given that catostomids are tetraploids (Uyeno & Smith, 1972), we cloned nuclear PCR products to ensure orthologous sequences were used in subsequent phylogenetic reconstructions. Purified PCR products were cloned using the TOPO TA Cloning® Kit (Invitrogen Corp., Carlsbad, CA, USA) with TOP10 chemically competent or electrocompetent cells. Positive colonies were chosen randomly and cultured, and then plasmids preps were purified using the QIAprep Spin Miniprep Kit (QIAGEN, Valencia, CA, USA) and sequenced in both directions using universal M13 primers. We sequenced all genes in both directions on an Applied Biosystems 3100 Genetic Analyzer using ABI PRISM BigDye Terminator v2.0 or v3.0 Cycle Sequencing Kits (Applied Biosystems, Foster City, CA, USA).

We edited sequence chromatographs, assembled sequence contigs, and created final DNA alignments using Geneious v5.4 (Kearse et al., 2012). We translated all gene sequences into amino acid sequences to check alignments for stop codons or elevated nonsynonymous substitution numbers, because these signatures can indicate the presence of nuclear mtDNA gene copies, or “NUMTs.” mtDNA and IRBP sequences aligned straightforwardly “by-eye” in Geneious. However, we aligned nuclear RPS7 sequences, and nuclear genes from other studies listed below, in MAFFT v6.850 (Katoh & Toh, 2008) using the local-pair FFTS algorithm with MAXITERATE = 50.

Dataset construction, model selection, and Bayesian phylogenetic analyses

We collated seven datasets for our analyses that we describe here, and which correspond sequentially to datasets listed in Table 3 and trees shown in Figs. 2A–2G (see Results). (1) The “concatenated mtDNA” dataset consisted of mtDNA cytb and ND2 sequences for 126 tips (121 sucker samples and five outgroup samples) plus 58 mtDNA cytochrome oxidase subunit 1 (cox1) gene sequences from GenBank. (2) A “four-locus” DNA dataset with the same 126 tips but adding the following nuclear data to the concatenated mtDNA dataset: 44 GHI sequences from Clements, Bart & Hurley (2012) and GenBank, plus 113 IRBP sequences and 52 RPS7 sequences from our study. (3) We reanalyzed a morphological dataset first presented by Smith (1992), containing 123 morphological characters for 64 taxa including 62 sucker taxa and two outgroup taxa. This matrix contained two extinct taxa: the historically extinct †Moxostoma lacerum, and †Amyzon, an Eocene–Oligocene genus of fossil suckers composed of five valid species known from British Columbia, Washington, Nevada, and Wyoming (four species), and Jilin province, China (Bruner, 1991; Chang et al., 2001; Smith, 1992; Appendix S1). (4) The fourth “mtDNA + morphology” dataset combined datasets 1 and 3. (5) Our “total-evidence” dataset contained all morphological and molecular characters analyzed in this study, for 85 taxa. Two final nDNA datasets were: (6) a “nuclear IRBP” alignment for the only nuclear gene with complete sampling, and (7) a “concatenated nDNA” alignment containing all three nuclear loci. Overall, the morphology dataset was analyzed alone (dataset 3; see Results Fig. 2C), in concert with mtDNA genes (dataset 4; see Results Fig. 2D), and combined with the full DNA sequence dataset in total-evidence analyses of dataset 5 (e.g. see Results Fig. 2E).

Table 3 Characteristics of each of the seven morphological and molecular datasets analyzed in this study.

Dataset	n (No. ingroup tips)	No. characters (missing data % of total)	No. parsimony informative characters (% total)	Data subsets	
1	2	3	4	5	6	7	†M	
Concatenated mtDNA	126 (121)	2,836 bp (11.9%)	1,216 (42.9%)	+	+	+	−	−	−	−	−	
Four-locus	85 (80)	5,925 bp (23.1%)	1,893 (31.9%)	+	+	+	+	+	+	+	−	
Morphology	64 (62)	123 (11.7)	120 (97.6%)	−	−	−	−	−	−	−	+	
mtDNA + morphology	85 (80)	2,959 (11.9%)	1,336 (45.2%)	+	+	+	−	−	−	−	+	
Total-evidence	85 (80)	6,048 (23.3%)	2,013 (33.3%)	+	+	+	+	+	+	+	+	
Nuclear IRBP	113 (108)	839 bp (0.3%)	210 (25.0%)	−	−	−	+	+	+	−	−	
Concatenated nDNA	113 (108)	3,089 bp (36.4%)	677 (21.9%)	−	−	−	+	+	+	+	−	
Notes:

Data subsets: 1 = mtDNA 1st codon sites; 2 = mtDNA 2nd codon sites; 3 = mtDNA 3rd codon sites; 4 = IRBP 1st codon sites; 5 = IRBP 2nd codon sites; 6 = IRBP 3rd codon sites; 7 = GHI + RPS7 sites, combined; M = morphological characters; “+” symbols indicate that a given subset was included in the dataset while “−” symbols indicate it was not.

† During BEAST analyses, but not MrBayes analyses, morphological characters were subdivided into various subsets described in the text and in the Mendeley-deposited input files (DOI: 10.17632/trw6sb4v7w.2).

Figure 2 Consensus topologies from MrBayes (Ronquist et al., 2012) analyses of the concatenated mtDNA (A), four-locus (B), morphology (C), mtDNA + morphology (D), total-evidence (E), concatenated nDNA (F), and nuclear IRBP (G) datasets.

Subfamily and tribe names and colors shown in panel (A) are followed throughout; in panels (B–G), text labels highlight positions of different taxa, including subfamilies (Myx., Myxocyprininae), tribe Catostomini, and Pantosteus, and dagger symbols indicate extinct taxa. Filled black circles indicate nodes with significant Bayesian posterior probabilities (BPP ≥0.95), and gray circles indicate near-significant nodes (BPP = 0.80–0.94). Scale bars are given in units of substitutions/site.

We selected the most appropriate partitioning schemes and models of sequence evolution for each of the DNA data “subsets” or “blocks” used in our phylogenetic analyses in PartitionFinder v1.1.1 (Lanfear et al., 2012), which included codon positions of the mtDNA and IRBP alignments, plus the GHI and RPS7 genes. We ran PartitionFinder simultaneously on all initial DNA subsets using the greedy heuristic search algorithm, which we set to conduct model comparisons to determine the “best-fit” partitioning scheme based on the Bayesian information criterion (BIC). PartitionFinder relies heavily on PhyML, which we set to link branch lengths, search 56 substitution models, and estimate the base frequencies, proportion of invariant sites (I), and the gamma shape distribution (Γ) of each model using maximum-likelihood.

We performed partitioned Bayesian phylogenetic analyses on all seven datasets in MrBayes 3.2.2 (Ronquist et al., 2012). Analyzing different taxon and character combinations allowed us to evaluate the effect of different data types, and of including morphological data and extinct taxa, on phylogenetic analyses of suckers. During molecular analyses, we specified partitioning schemes and best-fit models of sequence evolution selected in PartitionFinder, except where the selected model was not implemented in MrBayes we used the next most closely related model in the general time-reversible (GTR) family of models. Given that rate variation among morphological characters can confound phylogenetic branch lengths, applying Γ-distributed rates can greatly improve models of morphological evolution (Clarke & Middleton, 2008). Thus, for the morphological analysis, we specified Lewis’s (2001) Markov variable (Mkv) model with Γ-distributed rate heterogeneity and left characters unordered in state polarity (default). For each dataset, we conducted three independent MrBayes runs of eight chains, each with a Markov chain Monte Carlo (MCMC) chain length of 50 million generations. We diagnosed run convergence using the potential scale reduction factor, which should approach values of 1 when stationarity has been reached (Ronquist et al., 2012).

Bayesian total-evidence dating and relaxed-clock partitioning

We estimated divergence times as times to the most recent common ancestor (tMRCAs) through Bayesian relaxed molecular clock analyses of the total-evidence dataset in BEAST v2.4.5 (Bouckaert et al., 2014). For our tree prior, we employed the fossilized birth-death (FBD) process model (Gavryushkina et al., 2014; Heath, Huelsenbeck & Stadler, 2014) as modified by Gavryushkina et al. (2017), because this model accommodates total-evidence datasets, avoids the need for arbitrary calibration densities, and accommodates all fossils available for a group, and not merely ad hoc selections. In addition to 83 extant ingroup and outgroup samples, 19 extinct sucker taxa and their available minimum ages from the fossil record were included in the BEAST analyses based on evidence provided in Appendix S1. We calculated the approximate sampling proportion for extant lineages (ρ) as 0.97 and set the FBD time of origin prior (tor) to the Cypriniformes “Root” node calibration discussed in Appendix S1. Morphological characters were partitioned into groups having the same number of states, and each partition was assigned an Mkv model (Lewis, 2001; conditioning on the use of only variable characters) with Γ-distributed rate variation (“partitioned mode” in Gavryushkina et al., 2017). We partitioned the DNA data and set site models according to the best scheme identified in PartitionFinder. Other authors have employed FBD models on a fixed topology (Heath, Huelsenbeck & Stadler, 2014) or simultaneously estimated topology and FBD parameters (Gavryushkina et al., 2017). We allowed most nodes, including subfamily relationships, to change freely. However, we constrained the Catostomini and Moxostomatini crown groups to be monophyletic, consistent with our MrBayes results and hypotheses of previous studies (see Results and Discussion sections). Posterior nodal support can be markedly weakened when fossils have unscored characters (Gavryushkina et al., 2017) or lack data. To avoid spurious relationships arising from the large proportion of fossil Ictiobinae taxa (57%) and Myxocyprininae taxa (50%) lacking data, samples of Carpiodes, Ictiobus, Ictiobinae, and Myxocyprininae were each constrained to be monophyletic, consistent with our other phylogenetic results.

According to Ho & Lanfear (2010), implementing multiple relaxed-clock models for different data subsets in a data-partitioning scheme can provide a more biologically realistic way to model among-lineage rate variation and improves the fit of relaxed-clock models to the data. Using a relaxed-clock partitioning scheme may also yield more precise date estimates with narrower credible intervals (Ho & Lanfear, 2010). We statistically tested whether allocating separate uncorrelated lognormal relaxed clocks to different DNA data subsets, through “relaxed-clock partitioning,” yielded divergence time estimation models that provided a better fit to the data than assuming a single model of branch-specific rates across all data subsets. First, we estimated divergence dates and marginal likelihood scores for each of 12 different relaxed-clock partitioning models (M1–M12) with one to eight clocks (see Results and Discussion). Second, we estimated Bayes factors and conducted Bayesian model selection to identify the best-supported model.

We ran five replicate searches of each model in BEAST (MCMC = 2 × 107 generations, sampling every 4,000) using the “BEASTRunner.sh” script in PIrANHA (Bagley, 2017). We then estimated log-marginal likelihoods for each model by conducting path sampling (PS) (Baele et al., 2012) for 100 steps (106 generations each), while specifying a ∼B(0.3, 1) distribution for spacing the path steps (Xie et al., 2011). We calculated 2loge(B10) Bayes factors from the log-marginal likelihoods and evaluated “weight of evidence” of the models according to criteria in Kass & Raftery (1995). We took posterior distributions from the best-supported model as our best estimates of the time tree and divergence dates for Catostomidae lineages. We summarized parameters from the best-supported model, and ensured convergence and adequate effective sample sizes (ESS >100–200), using Tracer v1.5 (Rambaut & Drummond, 2014). We calculated a maximum clade credibility tree annotated with mean node ages from 5,000 post-burn-in trees in TreeAnnotator v2.4.5.

Using phylogenetic informativeness profiles to exclude problematic characters

Among other factors influencing phylogenetic inference, such as sampling effects on branch lengths or nodal support values (Heath, Hedtke & Hillis, 2008; Pyron, 2011), the varying informativeness of different character sets can substantially and adversely affect phylogenetic divergence dating results (Dornburg et al., 2014). Fortunately, recent methods for quantifying and visualizing “phylogenetic informativeness” (PI) profiles of character sets through time (Townsend, 2007; Townsend, Su & Tekle, 2012) provide a framework for identifying and excluding problematic character sets (Dornburg et al., 2014). One common pitfall is the use of characters whose profiles exhibit a decline in informativeness towards the root. As noted by Townsend & Leuenberger (2011), this decline marks a “rain shadow of noise,” with the corresponding dataset losing phylogenetic informativeness due to an increase in predicted homoplasy. In turn, homoplasious loci or character sets, but especially mtDNA datasets exhibiting high saturation or rootward declines in PI, have been shown to mislead global branch length values during divergence time estimation (Brandley et al., 2011; Dornburg et al., 2014).

To identify and exclude potentially problematic character sets in our database, and to evaluate whether support vs instability of subfamily relationships correlated to statistical power to revolve branching order, we evaluated the Townsend (2007) PI and resolution probabilities of each character set through time using PhyDesign (López-Giráldez & Townsend, 2011). Estimating PI requires prior information on evolutionary-genetic rates and phylogeny. Thus, we estimated rates for DNA characters and morphological characters using HyPhy (Pond, Frost & Muse, 2005) and BayesTraits (Pagel & Meade, 2014), respectively, and ran analyses along the BEAST time tree from the best-supported clock-partitioning model (see Results and Discussion). We estimated net PI for all eight data subsets in Table 3 during the “subfamily divergence epoch” spanning branches leading to Myxocyprininae, Cycleptinae, Ictiobinae, and Catostominae. These analyses permitted broad comparisons of signal in the mitochondrial vs nuclear data, and molecular vs morphological character sets. We excluded character sets that exhibited steep declines in PI towards the root of our phylogeny from the final divergence time results presented below. We compared three resolution probabilities, including probability correct, probability polytomy, and probability incorrect or “phylogenetic noise” (Eqs. 11–13 in Townsend, Su & Tekle, 2012) for the excluded subsets vs other molecular subsets over the subfamily divergence epoch. We also evaluated sensitivity of resolution probability approximations to varying the internode time length (t0) parameter, by recalculating over 10 t0 values representing declining fractions of the epoch.

Results

Dataset characteristics and DNA substitution models

Our final data matrices ranged in size from 123 characters in the morphological dataset to 6,048 molecular and morphological characters in the total-evidence dataset (Table 3). Proportions of missing data and parsimony informative characters ranged from 0.3% to 36.4% and from 21.9% to 97.6%, respectively, across datasets (Table 3). PartitionFinder identified seven unique DNA sequence subsets (scheme BIC = 141854.71523), and the best-fit DNA substitution model for each subset is listed in Table S1. Morphological character subsets (in subset M) were assigned Mkv+Γ models, as described above. None of the mtDNA genes sequenced in this study showed signs of NUMTs, and we found no indels in the IRBP sequences; however, GHI and RPS7 genes aligned with ∼32 and ∼16 ingroup indels/gaps, respectively. We archived our sequence alignments and phylogenetic tree results in Mendeley Data (DOI: 10.17632/trw6sb4v7w.2).

Phylogenetic relationships among sucker subfamilies

We placed subfamilies Myxocyprininae and Cycleptinae as sister lineages in most trees, but with higher posterior support (Bayesian posterior probability (BPP) = 0.74–0.99) in runs based largely on mtDNA-encoded genes (Figs. 2A and 2D; Figs. S1 and S2) and weak BPP support (≤0.69) in other runs (Figs. 2C, 2E, 3; Fig. S3). Frequently, when this pattern was obtained, Myxocyprininae + Cycleptinae was resolved as sister to a monophyletic Ictiobinae with variable posterior support (BPP = 0.69–0.98; Figs. 2, 3; Figs. S1 and S2). This “subfamily pattern 1,” with ((Myxocyprininae, Cycleptinae), Ictiobinae), departs markedly from the placement of Ictiobinae sister to Cycleptinae (sometimes including Myxocyprinus) + Catostominae in previous analyses of morphology (Smith, 1992) and molecular data (Doosey et al., 2010; Harris & Mayden, 2001; Mayden et al., 2008; Saitoh et al., 2006). By contrast, several trees agreed in placing Myxocyprininae as sister to all other sucker lineages, which agrees with previous mtDNA results presented in Harris & Mayden (2001). This “subfamily pattern 2” relationship was strongly supported with BPP = 1 in our total-evidence consensus topology from BEAST (Fig. 4) and resolved with low support in the four-locus topology (Fig. 2B; Fig. S4). Distinguishing between these two conflicting sets of subfamily relationships is difficult, because each is supported by molecular and total-evidence topologies herein and agrees with at least one previous molecular study. To objectively determine the arrangement of these subfamilies with the greatest weight of evidence conditional on our total-evidence dataset, we compared subfamily patterns 1 and 2 using Bayes factors. We ran MrBayes as described above, except employing topological constraints set to subfamily pattern 2, and then used stepping-stone sampling (Xie et al., 2011; Baele et al., 2012) to estimate the log-marginal likelihoods of the models, from which Bayes factor tests were conducted through comparisons to the unconstrained model matching subfamily pattern 1. Conducting 50,000 generations of stepping-stone sampling (sampling every 2,500 generations) during each of 50 steps produced a total of 250,000 MCMC generations for marginal likelihood estimation. The subfamily pattern 2 model constraining Myxocyprininae as sister to all other suckers had a higher log-marginal likelihood score (−68390.83) than the unconstrained subfamily pattern 1 model (−68910.88), and a 2loge(B10) Bayes factor of −1040.10 provided definitive weight of evidence against the unconstrained model.

Figure 3 Phylogeny of suckers inferred from Bayesian analysis of the combined mtDNA (cytb, cox1, ND2), nDNA (IRBP, GHI, and RPS7), and morphological data (123 characters) in the total-evidence dataset.

Within tip labels, dagger symbols indicate extinct taxa, and museum (voucher) or field numbers are followed by the number of the individual sequenced (Table 1). Nodes are labeled with Bayesian posterior probability support values above 0.50. Scale bars are in units of substitutions/site.

Figure 4 Time-calibrated phylogeny of Catostomidae derived from the best-supported Bayesian total-evidence dating model identified in Table 4.

In parentheses beside subfamily and tribe names, numbers of extinct or fossil tips are presented out of total sample size for the corresponding clade. Along nodes, filled black circles indicate significant posterior support (BPP ≥0.95), gray circles indicate near-significant support (BPP = 0.80–0.94), and horizontal node bars show 95% highest posterior densities of age estimates. The inset table shows Bayesian posterior age estimates given as mean tMRCAs, with their 95% highest posterior densities (HPDs; credible intervals), in units of millions of years ago (Ma). Geological epoch abbreviations: Paleo., Paleocene; Oligo., Oligocene; Q., Quaternary; Pl., Pliocene; P., Pleistocene.

Relationships of early-diverging sucker genera

We found that the early-diverging sucker genera Myxocyprinus, Cycleptus, Carpiodes, and Ictiobus formed well-supported clades in most analyses. However, Ictiobus relationships were resolved in a polytomy in our analysis of the concatenated mtDNA dataset (Fig. S1). In several other trees, including those based on our total-evidence and mtDNA + morphology datasets (Fig. 3; Fig. S2), the Ictiobus clade received weak BPP support and eventually collapses into a paraphyletic grade. In the analysis of the four-locus dataset with higher numerical sampling in this clade, Ictiobus bubalus and I. niger relationships had high BPP support (BPP = 0.94–1) but were para-/polyphyletic, leaving their relationship to I. cyprinellus uncertain (Fig. S4).

Phylogenetic relationships among the Catostominae

Within the largest sucker subfamily, Catostominae, we consistently resolved clades with the tribes Thoburniini + Moxostomatini and Erimyzonini + Catostomini across analyses. Our more robust, multilocus and total-evidence trees resolved these relationships with definitive support values of BPP = 0.99–1. Within the Thoburniini + Moxostomatini clade, we consistently inferred the genus Thoburnia to be paraphyletic, with Thoburnia atripinnis sister to a clade containing the three Hypentelium species (mostly BPP = 1). The sole exception to this was that our morphology tree resolved Thoburnia as monophyletic with T. atripinnis sister to all other Thoburnia with strong support (BPP = 1). Within the Hypentelium clade, we inferred an identical and strongly supported set of relationships of the form (Hypentelium roanokense, (H. etowanum, H. nigricans)) in the mtDNA, four-locus, nDNA, and total-evidence gene trees (Fig. 3; Figs. S1, S2, S4, and S5). We obtained the same set of relationships in our morphology analysis, but with weak (BPP = 0.69) support for the H. etowanum–H. nigricans node (Fig. S3). We resolved Moxostoma as monophyletic with BPP = 0.89–1, except for a paraphyletic pattern in the morphology consensus tree. Within the Erimyzonini, Erimyzon was monophyletic (BPP = 1) and sister to Minytrema (e.g. Fig. 3). None of the molecular or total-evidence topologies we inferred resolved Catostomus as monophyletic relative to Chasmistes, Deltistes, or Xyrauchen. Here, yet again, results from the morphology tree departed from our other results, failing to resolve relationships among these or virtually any other catostomine lineages with strong support (Fig. 1C; Fig. S3). As a result, we do not discuss the morphology consensus topology further in this section.

Relationships within the Erimyzonini and Catostomini were similar to those in previous molecular studies (Harris et al., 2002; Doosey et al., 2010). Within Erimyzon, our results placed E. sucetta rather than Erimyzon oblongus as sister to E. tenuis. Within Catostomini, we consistently resolved nine well-supported major clades within Catostominae (e.g. Fig. 3; Figs. S1 and S4). Although relationships among these clades received varying posterior support, the species groups we identified were highly supported in multiple analyses and provide more tenable phylogenetic hypotheses than previously proposed for this tribe (Smith, 1992; Smith et al., 2002). “Clade 4” corresponded to a monophyletic Erimyzonini, while clades 5 through 9 included various Catostomini subclades composed largely of Catostomus samples. For conciseness, we provide an in-depth assessment of relationships only within “Clade 5,” whose results have the most important phylogenetic and taxonomic implications. More granular presentation and discussion of relationships within and among Catostomini clades 1–3 and 5–9 is provided at the end of Appendix S1. “Clade 5” corresponded mostly to the genus Pantosteus (Unmack et al., 2014) and was sister to the remaining Catostomini. Within Clade 5, we consistently resolved Pantosteus nebuliferus + P. plebeius as sister to a clade containing all remaining Pantosteus, with (P. platyrhynchus, (P. santaanae, (P. clarkii, (P. d. discobolus, P. d. yarrowi)))) (e.g. Fig. 3). Alternative topologies inferred for this clade involved rearrangements placing P. santaanae sister to P. clarkii, but with non-significant posterior support (e.g. Figs. S1 and S4).

Bayesian total-evidence dating and relaxed-clock partitioning

Bayes factor comparisons of 12 clock-partitioning models showed that removing mtDNA 1st and 3rd codon sites deemed to be problematic during PI profiling (see below) progressively improved model log-marginal likelihoods and posterior evidence (Table 4). For example, codon-partitioned relaxed-clock models were overwhelmingly supported over simpler models allocating a single relaxed clock to all data subsets, or all DNA subsets. The most complex model allocating relaxed clocks to each data subset but including only mtDNA 2nd positions, M12, was decisively supported as the best model. Compared with this model, other subset and clock schemes produced negative improvements to the model, indicated by negative log Bayes factors; however, the second best model separated DNA from morphological data subsets, consistent with a decoupling of molecular rates from morphological rates (Table 4). Independent runs of the best-supported BEAST model achieved ESS scores of >100–200 for all parameters and converged on similar phylogeny and parameter estimates including mean and 95% highest posterior densities (HPDs; i.e., credible intervals) for sucker tMRCAs. For example, the posterior ESS of the best model was 463 and that for the FBD model was 134. The only exception to this was the tMRCA for Moxostomatini, which received ESS scores between 72 and 89. Clocks on different data subsets exhibited substantial among-lineage rate heterogeneity, with posterior means and 95% HPD intervals of the “ucldStdev” (uncorrelated lognormal relaxed clock standard deviation) and “coefficientOfVariation” (coefficient of variation of branch-specific rates) statistics excluding zero (Fig. S7); thus, relaxed-clock models were warranted by the data.

Table 4 Bayes factor tests comparing twelve Bayesian relaxed-clock partitioning models applied to the total-evidence dataset.

Model	Mitochondrial DNA	Problematic sites (#)	Relaxed clocks (#)	Clock-partitioning	Log-marginal likelihood	2loge(B10) Bayes factor	
M1	1st, 2nd, and 3rd codon sites	2	1	[1, 2, 3, 4, 5, 6, 7, M]	−69,461.62	−89,832.36	
M2	1st, 2nd, and 3rd codon sites	2	2	[1, 2, 3, 4, 5, 6, 7] [M]	−69,078.47	−89,066.05	
M3	1st, 2nd, and 3rd codon sites	2	8	[1] [2] [3] [4] [5] [6] [7] [M]	−67,222.91	−85,354.94	
M4	1st + 2nd codon sites	1	1	[1, 2, 4, 5, 6, 7, M]	−38,876.51	−28,662.13	
M5	1st + 2nd codon sites	1	2	[1, 2, 4, 5, 6, 7] [M]	−38,385.92	−27,680.96	
M6	1st + 2nd codon sites	1	7	[1] [2] [4] [5] [6] [7] [M]	−37,616.86	−26,142.84	
M7	1st codon sites	1	1	[1, 4, 5, 6, 7, M]	−34,866.49	−20,625.92	
M8	1st codon sites	1	2	[1, 4, 5, 6, 7] [M]	−34,382.43	−19,673.97	
M9	1st codon sites	1	6	[1] [4] [5] [6] [7] [M]	−33,682.60	−18,237.36	
M10	2nd codon sites	0	1	[2, 4, 5, 6, 7, M]	−25,442.32	−1,793.75	
M11	2nd codon sites	0	2	[2, 4, 5, 6, 7] [M]	−25,138.71	−1,186.53	
M12	2nd codon sites	0	6	[2] [4] [5] [6] [7] [M]	−24,545.44	0	
Notes:

Models including only 3rd codon sites of the mtDNA matrix were ruled out, because those sites had the highest amounts of predicted homoplasy during PI profiling (Results and Discussion, Fig. 5), and excluding them yielded the single greatest improvement in log-marginal likelihoods (e.g. M1 vs M4). Numbers and letters in brackets correspond to data subsets defined in the text and Table 3. Log-marginal likelihood estimates were derived from path sampling, and Bayes factors were estimated for each model when compared against the best-supported model.

In the final total-evidence dating analysis, the mean posterior age estimate for the tMRCA of all suckers was 63.16 Ma in the Late Cretaceous, with credible intervals ranging from Late Cretaceous to the Paleocene–Eocene boundary (95% HPD [54.02, 74.6]; Fig. 4). The four sucker subfamilies had variable posterior age estimates ranging approximately an order of magnitude. The Cycleptinae had the youngest posterior age estimate of 5.07 Ma in the Pliocene (95% HPD [0.87, 10.23]). Following their Late Cretaceous origin based on a stem age corresponding to the MRCA of all suckers, the posterior tMRCA estimate for Myxocyprininae dated their diversification to 42.27 Ma in the early to mid-Eocene (95% HPD [39.67, 54.58]). Subsequently, Catostominae species diversified since an intermediate posterior age of 34.37 Ma near the Eocene–Oligocene boundary (95% HPD [25.54, 42.77]), and Ictiobinae species had the oldest posterior age estimate, dating to 49.69 Ma in the early Eocene (95% HPD [48.88, 52.52]). The catostomine tribes diverged approximately 29.87 Ma in the Oligocene (Catostomini + Erimyzonini) and 20.78 Ma in the early Miocene (Moxostomatini + Thoburniini). The genera Catostomus and Moxostoma, which correspond to tribes Catostomini and Moxostomatini, diversified since 17.65 and 15.25 Ma ago in the early-mid Miocene, respectively (Fig. 4).

Phylogenetic informativeness profiles

We evaluated potential impacts of phylogenetic signal on incongruent subfamily relationships by estimating resolution probabilities, over the subfamily divergence epoch (∼63.2–34.4 Ma) spanning the divergence of sucker subfamilies. Overall, mtDNA 1st and 3rd codon position data subsets exhibited among the highest PI values, but with distinct Miocene peaks followed by declining PI towards the root (Fig. 5). This suggested a prominent loss of evolutionary information due to homoplasy; therefore, we excluded these sites from final divergence dating analyses, in order to avoid potentially negative effects on the topology and time-calibrated branch lengths (Dornburg et al., 2014). All other molecular data subsets had substantial and relatively constant predicted PI decaying over Paleocene or Eocene to present (recent spikes are anomalies; Townsend, López-Giráldez & Friedman, 2008). Morphological characters had slightly higher signal than IRBP sites and exhibited stability before decaying 20 Ma to present (Fig. 5). In addition to nearly constant net PI (Fig. 5), the retained character subsets also had low probabilities of phylogenetic noise or polytomies, with notable increases in the probability of an incorrect topology only for internode distances less than ∼0.35(t0), or <10.0 million years (Myr) (Fig. S9).

Figure 5 Phylogenetic informativeness of the sucker datasets.

(A) BEAST chronogram from Fig. 4 used for site rate calculations, showing the three main subfamilies with incongruent relationships highlighted in red. Phylogenetic informativeness profiles matched to the chronogram time scale are shown for seven subsets of the four-locus dataset (B), and for the morphology dataset (C). Colored shading in (B) and (C) indicate areas integrated below each profile over the “subfamily divergence epoch” (∼63.2–34.4 Ma) containing branching relationships among the four subfamilies.

Discussion

Sucker phylogeny and incongruence of subfamily lineages

Our phylogenetic reconstructions of Catostomidae relationships are similar to several previous morphological and molecular studies. For example, others have hypothesized that Catostomidae is monophyletic in studies focusing on suckers (Doosey et al., 2010; Ferris & Whitt, 1978; Harris & Mayden, 2001; Smith, 1992; Fig. 1) and taxonomically broader analyses (Mayden et al., 2008; Saitoh et al., 2006). Our finding that the four currently recognized sucker subfamilies are monophyletic with definitive support also agrees with earlier phylogenetic studies based on morphology (Smith, 1992) and molecules (Chen & Mayden, 2012; Clements, Bart & Hurley, 2012; Doosey et al., 2010; Harris & Mayden, 2001; Harris et al., 2002; Sun et al., 2007). This is perhaps unsurprising, as we reanalyzed previous morphological and molecular datasets alongside new sequence data. Yet ours are the first results definitively supporting patterns of monophyly at the family and subfamily levels based on dense taxonomic sampling of mtDNA and nuclear gene sequences for all sucker genera and most species, plus total-evidence analyses, with BPP at or near 1 across datasets (Figs. 2, 3; Figs. S1–S6).

Previous molecular phylogenetic studies of higher-level sucker relationships have often encountered difficulty in resolving relationships among sucker subfamilies (Chen & Mayden, 2012; Doosey et al., 2010; Harris et al., 2002; Sun et al., 2007). Likewise, relationships among the Myxocyprininae, Cycleptinae, and Ictiobinae lineages were incongruent across analyses of different datasets (Figs. 2, 3; Figs. S1–S4), but with two main patterns that we deemed subfamily pattern 1, with the form ((Myxocyprininae, Cycleptinae), Ictiobinae), and subfamily pattern 2, with Myxocyprininae sister to all other sucker subfamilies. We distinguished between these two alternative hypotheses using a topological constraint test based on Bayes factors. The result yielded log-marginal likelihood estimates and Bayes factors giving definitive weight of evidence against the unconstrained subfamily pattern 1 model. Given this result, a placement of Myxocyprininae as sister to all other suckers seems most probable at this point, thus we favor the patterns of subfamily relationships in our four-locus and Bayesian total-evidence dating topologies that are consistent with this result. Nevertheless, the question still remains: What factors have likely influenced the difficulty of our study and previous studies to resolve phylogenetic relationships among sucker subfamily lineages? Overall, our phylogenetic informativeness analyses highlight two potential explanations for the observed incongruence in subfamily relationships across analyses. First, PI profiling identified the mtDNA 1st and 3rd codon position data subsets as problematic character sets likely compromised by homoplasy due to nucleotide saturation (Fig. 5); hence, we felt justified in excluding these sites from our final divergence dating analyses. Second, our results suggest that predicted phylogenetic noise of the combined datasets over the subfamily divergence epoch (Fig. S9) was most likely a limiting factor for resolving Cycleptinae as sister to Ictiobinae. Whereas internode distances for Catostominae and Myxocyprininae crown clades were generally longer, being ∼11 to 18 Myr in length, and associated with significant posterior support, that for Cycleptinae + Ictiobinae had a short internode distance of only 1 Myr (95% HPDs [0.01,9.1]) and non-significant posterior support in our time tree (Fig. 4). Together with the more frequent incongruence and lower support for Cycleptinae compared to Ictiobinae across our MrBayes topologies, this suggests that Cycleptinae acted as a “rogue taxon” switching positions on the tree (Aberer, Krompass & Stamatakis, 2012). We hypothesize that our BEAST total-evidence tree inferred subfamily relationships that were more consistent with Bayes factor tests, and had greater nodal support for early-diverging nodes, by limiting the rogue movements of Cycleptinae.

Monophyly of early-diverging sucker genera, and relationships within Ictiobus

The genera Myxocyprinus, Cycleptus, Carpiodes, and Ictiobus formed well-supported clades in our results. However, relationships among Ictiobus species were resolved in a polytomy or paraphyletic grade in several cases, limiting our resolution of this clade. These Ictiobus results disagree with previous mtDNA- or nDNA-based studies resolving relationships among Ictiobus species with strong maximum-likelihood bootstrap support (Doosey et al., 2010), and Smith’s (1992) hypothesis of relationships among four Ictiobus species based mainly on morphology. Given these findings, and that our current results fail to unquestionably place the Cycleptinae as sister to the Ictiobinae, drawing phylogenetic or taxonomic conclusions about Ictiobus species relationships would seem premature, and we recommend more in-depth analyses of these taxa.

Phylogenetic relationships among the Catostominae

Within the Catostominae, our multilocus and total-evidence results strongly supported sister relationships between Thoburniini + Moxostomatini, and between Erimyzonini + Catostomini. These findings agree well with previous molecular results (Harris et al., 2002; Clements, Bart & Hurley, 2012), except for mitochondrial trees in Doosey et al. (2010) showing the Erimyzonini as sister to all other clades within Catostominae. Interestingly, however, our catostomine relationships conflict with the analysis of Smith (1992), whose morphological data we re-analyzed. In Smith’s (1992) sucker phylogeny, the Erimyzonini is resolved as sister to a clade containing what are currently regarded as the Moxostomatini and Thoburniini (Harris & Mayden, 2001; Harris et al., 2002). Smith pointed out that this set of relationships was supported by >20 apomorphies that changed at the node representing the MRCA of these lineages in his parsimony tree. But this conclusion is only as sound as the phylogeny upon which character state transitions were mapped by Smith (1992), which, at this node and several other key nodes, is rejected by our mtDNA, nDNA, and multilocus trees, as well as total-evidence results from analyzing Smith’s data together with molecular datasets.

Regarding our Thoburniini + Moxostomatini clade, genus Thoburnia was inferred to be paraphyletic based on the placement of T. atripinnis sister to the Hypentelium clade, mostly with strong BPP support. Doosey et al. (2010) and Clements, Bart & Hurley (2012) obtained the same relationship for T. atripinnis. However, our morphology tree resolved Thoburnia as monophyletic, which is consistent with Smith’s (1992) original analysis of the morphological data we used, suggesting further data or analyses are needed to clarify these relationships. We consistently inferred H. roanokense as sister to a clade of H. etowanum + H. nigricans, across molecular, morphological, and total-evidence analyses, though with varying BPP (Fig. 3; Figs. S1, S2, S4, and S5). These results match relationships inferred by Buth (1980) using isozyme data reflecting variation at 40 putative loci. By contrast, our results conflict with Smith’s (1992) hypothesis, which resolved H. roanokense as sister to H. nigricans; however, this relationship was based on a single morphological character, dermethmoid spine shape. Taking this into consideration, the broad congruence between multiple data types, as well as our re-analysis of Smith’s (1992) data, suggests high confidence in the inference that H. roanokense is the earliest diverging lineage in the genus. Within our Moxostomatini clade, Moxostoma was monophyletic consistent with previous analyses (Clements, Bart & Hurley, 2012).

Relationships within the Erimyzonini and Catostomini were very similar to those in Harris et al. (2002) and consistent with Doosey et al. (2010), but they contradicted Smith (1992), especially by resolving relationships within Erimyzon while placing E. sucetta sister to E. tenuis. As in previous molecular results for Erimyzonini, Erimyzon was monophyletic and sister to Minytrema in our results. However, Catostomini genera were not generally obtained as monophyletic, and in no case was Catostomus monophyletic relative to Chasmistes, Deltistes, or Xyrauchen. Our molecular and total-evidence analyses consistently resolved nine well-supported major clades within Catostominae (e.g. Fig. 3; Figs. S1 and S2). Here, we focus on relationships within Clade 5, which corresponded to the former subgenus Pantosteus, which Unmack et al. (2014) recently elevated to genus. Smith (1966) recognized six species within Pantosteus: P. clarkii, Catostomus columbianus, P. discobolus, P. plebeius, P. platyrhynchus, and P. santaanae. We sampled all of these, including both subspecies of P. discobolus, but consistently inferred a polyphyletic Pantosteus, with P. nebuliferus (recognized as distinct from P. plebeius by Miller, Minckley & Norris, 2005; Nelson et al., 2004) falling within Clade 5 but Catostomus columbianus placed in Clade 8 (discussed below). A clade with P. nebuliferus + P. plebeius was frequently sister to all remaining Pantosteus (e.g. Fig. 3). Notwithstanding incongruent results among analyses in the two papers, the consensus of results from our study and those of Unmack et al. (2014) seems to lend strongest support to the former relationship, with P. santaanae sister to a clade containing P. clarkii and P. discobolus lineages. The polyphyly of Pantosteus and nebuliferus–plebeius sister relationship are concordant with the results of Doosey et al.’s (2010) analyses using RY-coding for third position mtDNA substitutions, although they inferred P. santaanae as sister to a clade containing other members of Pantosteus. These results also agree with mitochondrial and morphological analyses of Unmack et al. (2014). However, our results depart from Doosey et al. (2010) and agree better with Unmack et al. (2014) in strongly supporting a sister relationship between C. columbianus and C. tahoensis. That the morphological and molecular data analyzed herein support the monophyly and diagnosability of Pantosteus relative to Catostomus, without rendering Catostomus paraphyletic, strongly supports Unmack et al.’s (2014) decision to redefine Pantosteus to exclude C. columbianus. We note that this taxonomic arrangement is also consistent with studies on morphological and biochemical variation in western suckers (Koehn, 1969; Smith, 1992; Smith & Koehn, 1971). Also, C. columbianus has an open frontoparietal fontanelle, a key diagnostic character of this clade, whereas other Pantosteus species have the frontoparietal fontanelle closed or reduced to a narrow slit (Smith, 1966).

“Catostomus” polyphyly and introgressive hybridization

As noted above, Catostomus was never resolved in our study as monophyletic relative to Chasmistes, Deltistes, or Xyrauchen, and this result is concordant with phylogenetic results of Doosey et al. (2010) based on mtDNA ND4/ND5 sequences. Hybridization of Catostomus with Chasmistes, Deltistes, and Xyrauchen is well documented (Buth, Murphy & Ulmer, 1987; Markle, Cavalluzzi & Simon, 2005; Mock et al., 2006; Tranah & May, 2006), and may be related to the non-monophyly of Catostomus relative to Chasmistes and Deltistes. However, while Xyrauchen texanus has been documented to hybridize with Catostomus latipinnis and Catostomus insignis (Buth, Murphy & Ulmer, 1987; Hubbs & Miller, 1953), the majority of these reports evaluate hybridization between X. texanus and Catostomus latipinnis (Buth, Murphy & Ulmer, 1987). Samples of Xyrauchen used in this study originated from the Dexter National Fish Hatchery, which obtained the original hatchery stock of Razorback sucker from Lake Mohave, Arizona, where hybridization with Catostomus latipinnis has been documented but allozyme evidence indicates only low levels of introgression of Catostomus latipinnis with X. texanus (Buth, Murphy & Ulmer, 1987). Therefore, placement of Catostomus insignis sister to X. texanus here and by Doosey et al. (2010) suggests that introgression is not a factor in either study. As such, “Xyrauchen” embedded within Catostomus renders the latter polyphyletic. Even if hybridization-mediated introgression were considered as an ad hoc explanation of this pattern, this is difficult to distinguish from the more parsimonious hypothesis of common ancestry, and the available data do not demonstrate that any hybridization events among these taxa have corresponded to the Neogene–present timeframe of their divergences inferred by our time tree. Thus, we advocate the tentative placement of “Xyrauchen” into synonymy with Catostomus until additional fossil or molecular evidence rejects an inference of common ancestry in favor of Neogene hybridization of these taxa.

Bayesian total-evidence dating and relaxed-clock partitioning

In showing that the best clock-partitioned BEAST models excluded sites identified as problematic in our PI profiling analysis, the results of our Bayes factor clock-partitioning model comparisons bolster Ho & Lanfear’s (2010) recommendation that accounting for differences in substitution rates among data partitions through clock-partitioning is not only feasible but also improves phylogenetic divergence dating models. We believe that by employing a clock-partitioning scheme objectively chosen in this way allowed our final BEAST FBD analysis to more correctly estimate topology and rate variance among branches, and better handle rate heterogeneity of the retained characters. However, while the inclusion of fossil taxa in an FBD model in the final BEAST analysis certainly improved our divergence time estimates over what might be obtained using node calibration or tip-dating methods (Arcila et al., 2015; Gavryushkina et al., 2017), one limitation of this analysis was that nodal support was reduced within the Catostomini and Ictiobinae. This pattern was caused by rogue placements of fossil taxa lacking character data, which were constrained within these crown groups but made up ≥50% of tip sampling (Fig. 4). Still, this mainly caused misleading relationships and lowered nodal support within the Catostomini; after removing fossil taxa, relationships within Ictiobinae would be essentially identical to our preferred MrBayes topologies. After pruning extinct taxa, our time tree will provide a suitable basis for interrogating the comparative biogeography and evolution of all groups of suckers, except for patterns within Catostomini. One alternate way forward for researchers interested in using our results for comparative phylogenetics would be to convert one of our preferred topologies (Fig. 3; Fig. S2) to an ultrametric tree while constraining subfamily and tribal node ages to mean tMRCA estimates shown in Fig. 4.

A major goal of our study was to use our final total-evidence dating results to test hypotheses on the temporal diversification of suckers. Our divergence dating results (Fig. 4) generally agree with the fossil record but reject or confirm different molecular hypotheses about the temporal diversification of sucker subfamilies. Unsurprisingly, given our incorporation of all fossil sucker lineages in the paleontological literature under an FBD model (accounting for extant and fossil sampling levels), our BEAST results strongly support hypothesis H4 that Catostomidae lineages have diversified since ∼50 Ma in the Early Eocene, which is widely accepted as the minimum age of the origin of suckers based on stratigraphic information for the oldest sucker fossils (review and refs. in Appendix S1). Our results also support Sun et al.’s (2007) proposal, or our hypothesis H3, that catostomine lineages in the most speciose sucker subfamily went on to diversify since ∼20 Ma. Indeed, initial divergences and subsequent diversification of all four catostomine tribes has proceeded since around ∼34–17.6 Ma in the Eocene–Miocene, with 95% credible intervals ranging from 43 to 11.04 Ma (Fig. 4), and the tMRCAs for ∼81% (64/79) of extant species/lineages in our time tree (all catostomines) coincide with the last 20 Myr.

In contrast to hypotheses H3 and H4 discussed above, we reject two previous molecular hypotheses about the tempo of sucker evolution advanced by Sun et al. (2007). First, we reject hypothesis H1 because we infer that the Asian Myxocyprininae diverged from North American suckers during the Late Cretaceous, and the 95% credible intervals for this divergence do not overlap with their proposed ∼14 Ma Miocene date for the MRCA of Myxocyprinus and Cycleptus. Second, we reject H2 given that we infer an early Eocene origin for Ictiobinae, including the extinct †Amyzon and †Vasnetzovia ictiobine lineages, and this vastly predates Sun et al.’s (2007) proposed origin of the clade. Given Sun et al. (2007) produced divergence time estimates using only cytb divergences and a global molecular clock assuming a 2.0% Myr−1 pairwise rate for vertebrate mtDNA, there are too many methodological distinctions between our approach and theirs to pinpoint a single factor causing our results to contrast theirs so strongly. However, our more comprehensive and nuanced approach using Bayesian total-evidence dating not only allowed us to use a realistic FBD tree prior incorporating the speciation-extinction-fossilization sampling process (Gavryushkina et al., 2017), but also permitted estimation of evolutionary rates for each character subset analyzed. We inferred a slower rate of evolution for mtDNA 2nd position sites, 6.53 × 10−4 substitutions site−1 Myr−1, and in fact all DNA subsets (mean: 0.0055 substitutions site−1 Myr−1), than the global rate applied by Sun et al. (2007). This is partly due to our best model excluding the most variable mtDNA codon sites, but it appears that differences between our tMRCA estimates and theirs are not fully accounted for based on substitution rates alone. Nevertheless, our more appropriate modeling of the evolutionary processes producing variation in sucker DNA sequences and morphological characters, and extant and fossil taxon sampling, has allowed us to estimate older and undoubtedly more accurate divergence dates, especially for deeper nodes in the sucker phylogeny.

We infer divergence times for major sucker lineages that are conspicuously older than those recently estimated from multilocus analyses of other North American freshwater fish clades, including sunfishes and black basses (Centrarchidae; Near, Bolnick & Wainwright, 2005; Near et al., 2011), bullhead and madtom catfishes (Ictaluridae; Hardman & Hardman, 2008), and darters (Etheostomatinae; Near et al., 2011). Whereas the diversification of these major lineages has occurred since around the Eocene–Oligocene transition ∼34 Ma, a time of global cooling (Zachos et al., 2001), we infer an earlier Late Cretaceous–Eocene age for the onset of sucker subfamily divergences. While tip-dating approaches can lead to older divergence dates (O’Reilly, Dos Reis & Donoghue, 2015), we believe that this result accurately reflects the relatively longer timeframe of sucker evolution captured by the fossil record. This general timeframe for sucker evolution also correlates well to the Late Paleocene Thermal Maximum, a period of greater ambient and sea-surface temperatures, higher sea levels, and higher precipitation and humidity (Zachos et al., 2001). Sucker diversification thus appears to have initiated during a period of climate change and sea level rise, which may have facilitated the isolation of ancestral sucker populations. Our results also suggest that approximately 4–7 sucker genera may have been present in North America by the Oligocene, a period coinciding with the arrival of minnows in the family Cyprinidae on the continent based on the broader fossil record of North American teleost fishes (Cavender, 1986). Together with the molecular results from other studies above, this implies that the subsequent diversification of these genera, including at least Cycleptus, †Amyzon, Ictiobus, and Carpiodes as well as the speciose Catostominae, would have coincided with the diversification of most other major lineages of North American freshwater fishes.

Conclusions

We have presented the results of a phylogenetic analysis of Holarctic sucker fishes (family Catostomidae) drawing on the most comprehensive dataset to date and inferring, separately and jointly, the phylogeny and divergence times of suckers while including fossil taxa as tips. Our molecular and total-evidence results corroborated relationships hypothesized in previous molecular studies and yielded evidence in favor of some new hypotheses of relationships within and among subfamilies, for example, with Bayes factor support for Myxocyprininae sister to all other sucker lineages. Our study also highlights how using PI profiling to identify problematic character sets can subsequently improve or provide additional evidence for clock-partitioning scheme choice during Bayesian relaxed-clock divergence dating. Our divergence-dating results strongly supported the hypotheses that Catostomidae lineages have diversified since ∼50 Ma in the Early Eocene (Uyeno & Smith, 1972), and that tribes within the most speciose subfamily, Catostominae, have diversified since ∼20 Ma in the Eocene–Miocene (Smith, 1992; Sun et al., 2007). Moreover, we hypothesized that incongruent subfamily relationships were driven in part by problematic mtDNA 1st and 3rd codon sites, and by “rogue taxon” movements of Cycleptinae and fossil taxa, for example in our FBD process time tree. Our analysis could be extended to test this latter hypothesis using additional statistical analyses of rogue taxa (Aberer, Krompass & Stamatakis, 2012) and internode uncertainty (Zhou et al., 2017), and by additional resolution analyses employing Monte Carlo simulations and tests of their assumptions (Townsend, Su & Tekle, 2012), which were beyond the scope of the present study. Nevertheless, our results suggest that future studies of suckers will benefit from using PI profiles as a predictive tool to select loci for subsequent phylogenetic analyses (Dornburg et al., 2014).

Supplemental Information

Supplemental Information 1 Appendix S1. Catostomidae fossils, fossil and external calibration age priors, and additional discussion of phylogenetic relationships within the Catostominae.

Click here for additional data file.

Supplemental Information 2 Table S1. Data subsets and corresponding DNA substitution models selected in PartitionFinder for the four-locus dataset.

Model selection analyses using PartitionFinder v1.1.1 supported different best-fit models of DNA evolution for different data subsets (i.e. groups or ‘blocks’ of data, such as sites filtered by codon positions). When it was not possible to specify a given model in phylogenetic software, we used the next most closely related model in the GTR family of models. Symbols and abbreviations: Γ, gamma-distributed rate variation; bp, number of nucleotide base pairs; I, parameter representing proportion of invariable sites; n, sample size (numbers correspond to sequence alignment sizes, except for multilocus datasets the numbers in parentheses are sample sizes for each locus); no., subset number.

Click here for additional data file.

Supplemental Information 3 Fig. S1. MrBayes consensus tree derived from the concatenated mtDNA dataset.

Click here for additional data file.

Supplemental Information 4 Fig. S2. MrBayes consensus tree derived from the mtDNA + morphology dataset.

Click here for additional data file.

Supplemental Information 5 Fig. S3. MrBayes consensus tree derived from the morphology dataset.

Click here for additional data file.

Supplemental Information 6 Fig. S4. MrBayes consensus tree derived from the four-locus dataset.

Click here for additional data file.

Supplemental Information 7 Fig. S5. MrBayes consensus tree derived from the nuclear IRBP dataset.

Click here for additional data file.

Supplemental Information 8 Fig. S6. MrBayes consensus tree derived from the concatenated nDNA dataset.

Click here for additional data file.

Supplemental Information 9 Fig. S7. Posterior density plots of BEAST ‘ucldStdev’ and ‘coefficient of variation’ statistics demonstrating significant among-lineage rate heterogeneity in the total-evidence dataset.

Problematic mtDNA 1st and 3rd codon sites were excluded from the analysis.

Click here for additional data file.

Supplemental Information 10 Fig. S8. Variation in substitution rates among sites.

This figure plots variation in substitution rates among sites, based on posterior distributions of the gamma shape parameters (alpha estimates) assigned to different data subsets in BEAST (excluding problematic mtDNA 1stand 3rd codon position sites).

Click here for additional data file.

Supplemental Information 11 Fig. S9. Probabilities of different character sets inferring the correct topology, a polytomy, or the incorrect topology (‘phylogenetic noise’) over the period of sucker subfamily divergence.

The period investigated is the ‘subfamily divergence epoch’ (SDE; ∼63.16–34.37 Ma) in our time tree. The SDE defines an internode distance, t0, used in the probability approximations. Results are shown across relative t0 values reflecting t0 proportions declining from 100% (1) to 10% (0.1) along the x-axis.

Click here for additional data file.

We are grateful to the following colleagues for providing specimens used in this study: Tom Turner and Alexandra Snyder, Museum of Southwest Biology; Karen Mock and Brian Cardall, Utah State University; Mike Bessert, University of Nebraska Lincoln; Douglas Markle, Oregon State University; Dennis Shiozawa, Monte L. Bean Life Science Museum, Brigham Young University; Vicky Albert and Louis Bernatchez, Université Laval; Andrew Simons, Bell Museum of Natural History, University of Minnesota; and Morgan Raley, North Carolina State Museum of Natural Sciences. We thank Kenneth De Baets, Stephan Koblmüller, Guillermo Ortí, Rodolfo Pérez-Rodríguez, and three anonymous reviewers for valuable comments on earlier versions of this manuscript, and we thank Andrew Eckert for useful discussions of analyses and presentation. We also thank the Brigham Young University Fulton Supercomputing Lab for providing generous computational resources.

Additional Information and Declarations

Competing Interests

Author Contributions

Animal Ethics

DNA Deposition

Data Availability

The authors declare that they have no competing interests.

Justin C. Bagley conceived and designed the experiments, performed the experiments, analyzed the data, prepared figures and/or tables, authored or reviewed drafts of the paper, approved the final draft.

Richard L. Mayden conceived and designed the experiments, contributed reagents/materials/analysis tools, approved the final draft.

Phillip M. Harris conceived and designed the experiments, contributed reagents/materials/analysis tools, approved the final draft.

The following information was supplied relating to ethical approvals (i.e., approving body and any reference numbers):

All molecular laboratory work for this project conducted in the R. L. Mayden laboratory was approved under Saint Louis University Institutional Animal Care and Use Committee (IACUC) protocol #2467.

The following information was supplied regarding the deposition of DNA sequences:

GenBank accession numbers for sequences used in this study are listed in Table 2, and the raw sequence files are provided in the Mendeley Data accession (DOI: 10.17632/trw6sb4v7w.2).

The following information was supplied regarding data availability:

Bagley, Justin; Mayden, Richard; Harris, Phillip (2018), “Data for: Phylogeny and divergence times of suckers (Cypriniformes: Catostomidae) inferred from Bayesian total-evidence analyses of molecules, morphology, and fossils,” Mendeley Data, v2.

http://dx.doi.org/10.17632/trw6sb4v7w.2.

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
