# Peer review of "Phylogeny and divergence times of suckers (Cypriniformes: Catostomidae) inferred from Bayesian total-evidence analyses of molecules, morphology, and fossils"

_PeerJ, doi:10.7717/peerj.5168_

## Round 0.1 · original submission · Minor Revisions

I feel the manuscript on phylogeny and divergence time estimation in suckers is exceptionally well executed and well written. I agree with reviewer 1 that the model selection and sensitivity testing is well justified and adds robustness to the results. I also appreciated that you clearly spelled out the hypotheses you wanted to test using the divergence time estimation. I just had some minor points I would like you to address before publication. The main points are:

ESS scores: please report the statistic used to assess convergence (ESS scores > 100-200 are used as a standard: see comments by reviewer 1)

Datasets used: you speak about 7 datasets (line 152+), but only six are used (fifth dataset not mentioned; see comments by reviewer 2 and 3).

Terminology: you speak of multilocus analysis, but it might be more appropriate to talk about 2 loci (see comments by reviewer 3)

Please address the following points in addition to the suggestions of the reviewers:

Line 36: reject or confirm different molecular hypotheses – please be more specific on what is rejected and confirmed.

Line 52: not everyone might be familiar with operational taxonomic units or OTUs – it might therefore be good to cite a reference in this context

Line 76-77: I feel there might be more appropriate/crucial references in addition to Near et al. 2011, 2012 arguing for comprehensive tip sampling to understand tempo and mode of diversification

Line 88: Gavryusshkina et al. is not the first study to incorporate this – so it might might be appropriate to cite the studies who first used/introduced this approach (you cited them later on; e.g., Heath et al. 2014)

Line 287: it might be appropriate to spell out the abbreviation BPP the first time it is used.

Line 402: how did you decide that recent spikes are anomalies

Line 534: shouldn´t latippinis" be "latipinnis" and "insiginis" be "insignis"

Line 572: I would keep in the line referring the correspondence with fossil record (as opposed to the suggestion of reviewer 3). This is expectation, but still some divergence estimates might not align with the fossil record use to calibrate it, so it is worth to keep it in.

Line 596: I feel the use of “strongly” might be more appropriate than “starkly”

Line 610: we infer divergence times for major sucker lineages that are conspicuously older than those … estimated for … other … freshwater clades: could this also partially be related with the used method. O’Reilly et al. (2015) indicated that in some cases – tip-dating can lead to older ages.

Line 658: please also thank the additional reviewers ...

Suggested reference:

O’Reilly, J. E., dos Reis, M., & Donoghue, P. C. (2015). Dating tips for divergence-time estimation. Trends in Genetics, 31(11), 637-650.

Reviewer 1 ·

Basic reporting

Line 85: More references are needed for relaxed clocks, the uncorrelated log-normal/exponential clock are just two of many published relaxed clock methods.

Line 223: These aren’t “BEAST models”, they would be divergence time estimation models, or perhaps clock models.

I think it would also be commenting on the fact that the most complex clock partitioning scheme was the best supported one, and that the one that simply separates molecular and morphological rates was the next best supported one. This suggests that morphological rates are decoupled from molecular rates, which is an interesting result for certain.

Experimental design

Line 189: It is suspicious that the statistic used to assess convergence is varying throughout the manuscript. Even though this area is somewhat subjective, ESS scores >200 (or even 100) have emerged as a standard in Bayesian phylogenetics and their absence here could mean that ESS < 200 are being obtained. If this is the case then the reader/reviewer needs to know this. If it is not the case, then reporting the ESS score would reinforce the convergence of the MCMC. MrBayes output files “.p” can be opened in tracer, or R, and the ESS values for the parameter estimates can be obtained.

Validity of the findings

no comment

Additional comments

I feel that this manuscript is a great example of a divergence time analysis done well. The model selection and sensitivity testing is well justified and adds a robustness to the results. My only concerns relate to a lack of clarity on some very small points, clarifying these points would be worthwhile.

·

Basic reporting

• The article is written in a clear and technically corrected english.
• The background of the article is quite adequate and the literarure is apporpiately referenced.
• The structure presents an aceptable format.
• The article presents coherence and the findings resulted appropiate to assess the original hypotheses.

Experimental design

• This research is within the aims and scope of the journal
• Research question and hypotheses are clearly stated, and contribute to an importante part of knowledge in diffrenet topics, such as, sistematic and evolution of Catostomid fishes, and the application of relevant methological tools of analysis in sistematics and evolutionary topics.
• Does not present any technical or ethical objection.
• Methods section are described sufficiently.

Validity of the findings

• Findings obtained answer fully the research and hypotheses, and as was mentioned above, represents a relevant contribution within its line of research.
• Data sets and methodological analyses are appropiate and novel.
• Conclusion is fully linked to both, the research question and the results supported.

Additional comments

I recommended the acceptance of the manuscrit “Phylogeny and divergence times of suckers (Cypriniformes: Catostomidae) inferred from Bayesian total-evidence analyses of molecules, morphology, and fossils”, since to is a relevant contribution for the systematics and evolution of the Catostomidae. In addition, presents the application and validity of a novel methodology.

·

Basic reporting

This manuscript on phylogeny and divergence times of suckers is very well written. It provides a nice overview of the problem and specifies particular hypotheses to be tested in this study.

Experimental design

This research is clearly within the scope of the Journal. The Research questions are well defined and fill not only an important knowledge gap on the phylogeny and temporal framework of diversification of suckers, but moreover represents an exceptional showcase on how to perform a proper phylogenetic analysis plus divergence time inference and gives practical adives on how to optimizse such analyses. The methods are descibed detail so that readers will be able to follow their procedures and apply them to their own data.

Validity of the findings

This is an important study with great Impact in the field and as far as I can judge, the findings are based upon solid and statistically sound data. The conclusions are well stated and founded, linked to the original Research questions/hypotheses stated in the introduction.,

Additional comments

This is a very nice and well written study that I enoyed reading. Congratulation to a great piece of work. I only have a few very minor comments that will be easy to implement in a revised version:
1. introduction, lines 68 & 72: I would refrain from calling datasets/analysis based on just two loci "multilocus datasets/analyses". Exlplictely state that these analyses are based on 2 loci, but don't uses the word "mulitlocus" here.
2. M&M, lines 152+: Seven datasets? I only Count six (at least six are described in this paragraph). I cannot find a dataset 5.
3. Results, line 277: "0.3-36%" to "21.9-97.6%"? Shouldn't this read "0.3-36%" AND "21.9-97.6%"?
4. Results, line 363: ???? it appears that something's missing here after "(see below)". I'm not a native English Speaker, but to me this sentence, as it is, does not make much sense.
5. Results on phylogenetic informativeness profiles: Very interesting and unexpected (at least to me) that you include the 2nd mtDNA positions and not the 1st as well. I would expect (maybe I'm a bit naive) that These two behave similarly.
6. Discussion, line 534: Change "latippinis" to "latipinnis" and "insiginis" to "insignis"
7. Discussion, line 572: I think you can delete this sentence as it's clear that your results must agree with the fossil record (as you use the fossils to calibrate your tree).
8. Discussion, line 603: inflated mtDNA substituion rates in Sun et al. (2007)? Where's your evidence for that? Unless I've overlooked it, you only discuss substituion rates of your 2nd mtDNA positions in this context, and comparing rates of 2nd positions to rates of entire genes doesn not make much sense here.

---

## Round 0.2 · Minor Revisions

Thank you for addressing our suggestions, implementing most of them (when they made sense to you) and explaining your approach pertaining to ESS scores in greater detail. Your paper is as good as accepted.

I just had some minor suggestions i would like to take care of before publication. I feel it would make it easier to follow your paper and discussion if you refer to Figure 2 in the text when discussing the datasets and referring the same abbreviations used to refer to them subsequently (A-G).

More specifically, i would refer to "(see Fig. 2)" on line 154 after "analyses" and mention the particularly letter (A-G) pertaining to each particular dataset when they are discussed here (see annotated pdf). I would also mention here how these datasets were combined for completeness sake (e.g., morphology with mtDNA (D) and in the total evidence approach (E)).

---

## Round 0.3 · accepted · Accept

Thank you for referring to Figure 2 in the text when discussing the datasets and using the same designations to refer to them subsequently. This makes the manuscript even easier to follow. Thank you for also taking this opportunity to change the lines referring to “section 2.2" of your older manuscript which is no longer used.

#

---

## Author Rebuttal · Round 0.3

12 June 2018

Justin C. Bagley, Ph.D.
Department of Biology
Virginia Commonwealth University
Richmond, VA 23284-2012

Kenneth De Baets, Ph.D.
Academic Editor, *PeerJ*
Geozentrum Nordbayern
Friedrich-Alexander Universität Erlangen-Nürnberg
Erlangen, Germany

RE:  MS #26838, "Phylogeny and divergence times of suckers (Cypriniformes: Catostomidae) inferred from Bayesian total-evidence analyses of molecules, morphology, and fossils"

Dear Editor:

Thank you for your recent news concerning our manuscript. We were pleased to learn that you consider the manuscript essentially accepted for publication at *PeerJ*, pending minor revisions that you outlined in your decision letter.

      Please accept the corrected version of our manuscript, enclosed herein in files with tracked changes (with my edits highlighted in green, for ease of viewing) and without tracked changes. I was able to correct the manuscript as you indicated and fix all points raised, as illustrated by the brief itemized list below containing your suggestions followed by my responses highlighted in gray.

      Thanks very much to you and to the reviewers for suggesting ways of clarifying/improving the manuscript, and helping us bring it to this point. We look forward to seeing the manuscript published in open access format in your fine journal.

Sincerely,

Justin C. Bagley, Ph.D.
E-mail: jcbagley@vcu.edu

## Editor's Decision and Recommendations:

**MINOR REVISIONS**

Thank you for addressing our suggestions, implementing most of them (when they made sense to you) and explaining your approach pertaining to ESS scores in greater detail. Your paper is as good as accepted.

Thank you very much for this good news. We were happy to revise the manuscript and also believe that the paper is highly suitable for publication in *PeerJ*. We hope our edits below meet all of your expectations.

Editor point 1: I just had some minor suggestions i would like to take care of before publication. I feel it would make it easier to follow your paper and discussion if you refer to Figure 2 in the text when discussing the datasets and referring the same abbreviations used to refer to them subsequently (A-G).

Thanks for these specific suggestions. We fixed this by going back and rewriting this section of the Methods of the manuscript so that it refers to Figure 2A-G (referring the reader to the Results section), as well as Table 3 (which contains additional details on each of the seven datasets). We felt it could also increase readability to briefly note that the datasets are discussed in this section (and given in Table 3) in the same order as in the results Fig. 2. Specifically, we change the first sentence of the "Dataset construction…" section to read, **"We collated seven datasets for our analyses that we describe here, and which correspond sequentially to datasets listed in Table 3 and trees shown in Figs. 2A–G (see Results),"** at **Lines 156 to 157** of the revised draft with tracked changes.

In addition to making these changes consistent with the Editor's recommendations, we also noticed that one potential point of confusion might have been that dataset 4 was not outlined in this section, as you might expect. We fixed this by adding a brief description of this dataset to complete the list in this paragraph. This change occurs at **Lines 169 to 170** of the revised manuscript with tracked changes.

Editor point 2: More specifically, i would refer to "(see Fig. 2)" on line 154 after "analyses" and mention the particularly letter (A-G) pertaining to each particular dataset when they are discussed here (see annotated pdf). I would also mention here how these datasets were combined for completeness sake (e.g., morphology with mtDNA (D) and in the total evidence approach (E)).

As noted above, we now refer the reader to Fig. 2 at this point of the manuscript using letters A–G corresponding to the figure panels. We also followed your suggestion to state that the morphology data were analyzed in three different datasets (three separate analyses), which we felt was a good idea. Specifically, we appended to this section a brief sentence about morphology stating, **"Overall, the morphology dataset was analyzed alone (dataset 3; see Results Fig. 2C), in concert with mtDNA genes (dataset 4; see Results Fig. 2D), and combined with the full DNA sequence dataset in total-evidence analyses of dataset 5 (e.g. see Results Fig. 2E)."** We hope this completes this section and makes our description of the datasets used in the complex set of analyses we conducted more accessible to all readers.

## Additional Changes to the Revised Manuscript

We realized that two areas of the manuscript referred to "section 2.2", a carryover from a previous draft of the manuscript in which we had numbered the sections. We fixed this by replacing references to this

section with references to the ensuing text "below", at **Lines 116 and 153** of the revised manuscript with tracked changes.

Editor, thank you sincerely again for all of your help with this manuscript, and also please know that the annotated version of the manuscript that you provided was very helpful during this process.

Best regards,

JCB
Revised manuscript files enclosed